# High-Temperature Tribological Performance of Hard Multilayer TiN-AlTiN/nACo-CrN/AlCrN-AlCrO-AlTiCrN Coating Deposited on WC-Co Substrate

**Asad Alamgir** [1,*], **Maxim Yashin** [1], **Andrei Bogatov** [1], **Mart Viljus** [1], **Rainer Traksmaa** [1], **Jozef Sondor** [2], **Andreas Lümkemann** [3], **Fjodor Sergejev** [1] **and Vitali Podgursky** [1]

1 Department of Mechanical and Industrial Engineering, Tallinn University of Technology, Ehitajate tee 5, 19086 Tallinn, Estonia; mayash@ttu.ee (M.Y.); andrei.bogatov@mail.ru (A.B.); mart.viljus@taltech.ee (M.V.); rainer.traksmaa@taltech.ee (R.T.); fjodor.sergejev@taltech.ee (F.S.); vitali.podgurski@ttu.ee (V.P.)
2 LISS a.s., Coating center PLATIT, Dopravní 2603, 756 61 Rožnov pod Radhoštem, Czech Republic; j.sondor@liss.cz
3 PLATIT AG, Eichholzstrasse 9, CH-2545 Selzach, Switzerland; a.luemkemann@platit.com
* Correspondence: asad.shaikh@taltech.ee; Tel.: +372-58791387

**Abstract:** Mechanical and tribological properties of the hard-multilayer TiN-AlTiN/nACo-CrN/AlCrN-AlCrO-AlTiCrN coating deposited on WC-Co substrate were investigated. The sliding tests were carried out using ball-on-disc tribometer at room (25 °C) and high temperatures (600 and 800 °C) with $Al_2O_3$ balls as counterpart. Nano-scratch tests were performed at room temperature with a sphero-conical diamond indenter. The surface morphology and chemical composition were investigated with scanning electron microscopy (SEM), energy-dispersive spectrometry (EDS) and in-situ high-temperature X-ray diffraction (HT-XRD). The phase transition from fcc-$(Al,Cr)_2O_3$ into $\alpha$-$(Al,Cr)_2O_3$ was observed at about 800 °C. The results of the tribological tests depends on the temperature, the lowest apparent and real wear volumes were observed on the coating after the test at 800 °C along with the smallest coefficient of friction (COF). The plastic deformation of the coating was confirmed in sliding and nano-scratch tests. The nano-scratch tests revealed the dependence of COF value on the temperature of the sliding tests.

**Keywords:** multilayer coating; high temperature tribology; nano-scratch

## 1. Introduction

To achieve industrial demands, several advanced coating structures such as multi-component [1,2], nanocomposite [3], gradient [4,5] and multilayer structures [6,7] have been developed. Multilayer structure combines the advantages of component sublayers. The substitutional defects may form at the interface of adjacent layers in multilayered coatings, which results in the generation of strain energy preventing the movement of dislocations [8]. The stress concentrations act as crack inhibitors and thus improve the fracture resistance [9]. Therefore, multilayer structure is one of the widely used and convenient approach for practical applications of coating [10–12].

The nitride-based coatings are characterized by high hardness, high melting temperature, good fracture toughness, chemical and physical stability. The TiN coating has been extensively used to protect cutting tools, due to their superior mechanical properties. However, TiN failed because of coating surface oxidation occurring during the cutting process. The issue of surface oxidation was solved by the addition of Al in TiN, which leads to formation of thin protective alumina layer on the coating surface. The Al atoms substitute Ti atoms in TiN lattice. The lattice distortion leads to the increase of the hardness and thus improve the abrasive wear resistance of TiAlN [13,14]. The (nc-$Ti_{1-x}Al_xN$)/($\alpha$-$Si_3N_4$)

(nACo) coating is the nanocomposite coating of nanocrystalline nc-AlTiN grains embedded into amorphous α-Si$_3$N$_4$ matrix, which results in high hardness and good resistance to abrasive wear. The AlCrN coating was developed based on binary CrN coating [15]. The addition of Al into the CrN coating improves oxidation resistance, chemical stability and mechanical properties [16,17]. The deposition of AlCrO ((Al,Cr)$_2$O$_3$) improves the stability of the whole system at elevated temperature and reduces chemical affinity between the tool and workpiece in cutting process. Finally, the AlTiCrN coating was used as an adaptive layer [18,19]. Tribological adaptability plays a significant role in improving the tribological properties of hard coatings. The adaptability of the tribological system can be achieved, for instance, by increasing the chemical complexity of coating [20,21].

The substrate effects significantly on the tribological behavior of the coating, for instance, soft coating on hard substrate and the hard coating on soft substrate exhibit different tribological behaviors [22]. The load-carrying capacity is an ability to withstand pressure in the film/substrate system without the loss of the system functionality [23]. The film deflection phenomenon was investigated in our previous studies where the diamond films were deposited on the Si and single crystal diamond substrates [24–26]. The deflection of diamond film deposited on the Si substrate was observed in sliding tests. It is expected that during the cutting a high load experienced by coating. However, in spite of the importance of load-carrying capacity for the characterization of coated tools, this property remains insufficiently investigated, particularly, for multilayer hard coatings.

In the present study, the mechanical and tribological properties of hard-multilayers TiN-AlTiN/ nACo-CrN/AlCrN-AlCrO-AlTiCrN coating were investigated at elevated temperatures under ambient environment conditions.

## 2. Materials and Methods

The TiN-AlTiN/nACo-CrN/AlCrN-AlCrO-AlTiCrN coating was deposited on polished WC-Co substrates (Ø 13 mm, height 5 mm, R$_a$ = 0.05 μm) using physical vapor deposition (PVD) rotating cathodes arc Platit® π411 system. The Ti, AlTi (33 at.% Ti), AlCr (45 at.% Cr) and AlSi (18 at.% Si) cathodes were used for the deposition of coating. The TiN-AlTiN/nACo layers were deposited at 550 °C and CrN/AlCrN-AlCrO-AlTiCrN at 570 °C, the total thickness of coating was 3.6 μm. The thickness and hardness of layers are shown in Table 1. The thickness of coating was measured by the calo-test (BAQ®) method.

**Table 1.** Thickness and hardness of the coating layers deposited on WC-Co substrate.

| Substrate | Layers | Coating | Thickness (μm) | Nano Hardness (GPa) |
|---|---|---|---|---|
| **WC-Co** | 1 | TiN | 0.2 | 24–26 * |
| | 2 | AlTiN/nACo | 2.0 | 36–38 */39–41 * |
| | 3 | CrN/AlCrN | 0.6 | 21–23 */36–38 * |
| | 4 | (Al,Cr)$_2$O$_3$ (fcc and α phases) | 0.5 | 24–25 ** and 28 ** |
| | 5 | AlTiCrN | 0.3 | 36–38 * |

*—the hardness values adopted from Platit [27]. **—the hardness values adopted from Khatabi et al. [28].

The tribological behavior of coating was investigated using a ball-on-disc CERT tribometer (manufacturer Bruker® UMT-2, Billerica, MA, USA). The samples were subjected to tests at room (RT) (25 °C) and high temperatures (600 and 800 °C). All tests were performed under ambient conditions using Al$_2$O$_3$ (alumina) balls from REDHILL® with Ø 6 mm as the counterbody, with the normal load of 10 N. The speed of rotation was 300 rpm, the wear track radius was 4 mm, the total sliding distance was 5400 m and the duration of each test was 3 h.

The cross-section area of the wear scars was measured using mechanical profilometer (Mahr Perthometer®, Göttingen, Germany) at four different positions to calculate the wear volume.

The nano-scratch tests were performed with ultra-low drift nano-mechanical test system (NanoTest, Micro Materials Ltd.®, Wrexham, UK) with a sphero-conical diamond indenter with a tip radius of 5 μm. All tests were performed as multi-pass experiments (3-scans: topography–scratch–topography) that were subsequently analyzed in the NanoTest software to determine the on-load and residual depth. The total length of scratch tests was 300 μm and load applied for scratch after scanning the distance of 50 μm. The progressive load with loading rate of 5 mN/s was applied up to the reaching of the peak load, the details of the tests are presented in Table 2. The elastic/plastic behavior of coating, wear and coefficient of friction (COF) were the prime interest in nano-scratch tests.

**Table 2.** Nano-scratch tests parameters.

| Peak Load (mN) | Loading Rate, $dL/dt$ (mN/s) | Scan Speed, $dx/dt$ (μm/s) | $dL/dx$ (mN/μm) |
|---|---|---|---|
| 5, 10, 15, 20, 50, 75, 100, 150, 200, 250, 300, 400 | 5 | 2 | 2.5 |

The optical microscope and scanning electron microscope (SEM) were used for investigation of the surface morphology. The SEM images were taken by the (Hitachi® TM-1000 system, Tokyo, Japan) and the (Zeiss® EVO MA-15 system, Oberkochen, Germany) with LaB6 cathode in secondary electron mode, applying an accelerating voltage of 10–15 kV and 6.5–5.8 mm working distance. The chemical composition of the coating was investigated by energy dispersion X-ray spectroscopy (EDS) and high-temperature in-situ X-ray diffraction (HT-XRD) using a (Rigaku® Cu Kα radiation, Tokyo, Japan) with the heating rate of 20 °C/min and the temperature range between RT and 1200 °C.

## 3. Results and Discussion

Figure 1a shows the SEM image of pristine surface, spherical and shapeless microdroplets and surface cavities can be observed. The HT-XRD patterns collected between RT and 1200 °C with an interval of 100 °C are shown in Figure 1b. There is no significant difference between the patterns corresponding to as-deposited and annealed up to 700 °C coating, the XRD peaks correspond to the fcc-$(Al_{1-x}Cr_x)_{2+\delta}$ (fcc-$(Al,Cr)_2O_3$) [29], WC-Co [30] and cubic AlTiCrN (c-AlTiCrN) coating [28]. The formation of fcc-$(Al,Cr)_2O_3$ grown on top of c-AlTiCrN layer is in agreement with the study of Najafi H. et al. [29], namely, the XRD peak at 43.5° was found after the preparation of fcc-$(Al,Cr)_2O_3$ at 550 °C on the cubic CrN layer. The growth and mechanical properties of the fcc-$(Al,Cr)_2O_3$ oxide deposited by magnetron sputtering was investigated by Khatibi A. et al. [28], the nano hardness was 24–25 GPa. The XRD peaks corresponding to the α-$(Al,Cr)_2O_3$ phase can be observed under heating from 800 to 1200 °C (Figure 1b). The presence of α-$(Al,Cr)_2O_3$ indicates phase transition from fcc to α phase at 800 °C. The relative intensity of the peak at 43.5° decreases steadily at the higher temperatures and an increase of the intensity of the peaks corresponding to the α phase can be seen.

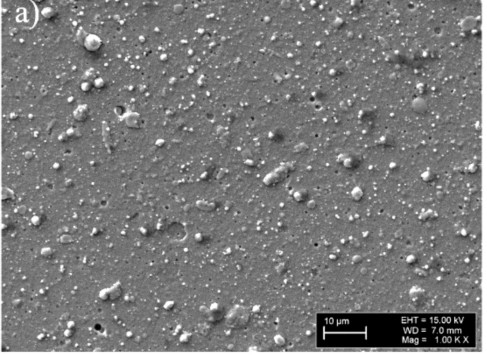 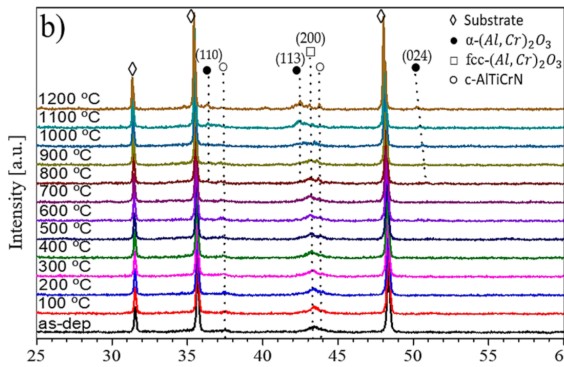

**Figure 1.** SEM image taken on pristine coating surface (**a**) and (**b**) HT-XRD patterns.

Figure 2 shows the COF versus cycles curves recorded during the sliding tests at RT, 600, and 800 °C. The steady-state regime without significant run-in-period was found from the beginning of the test at RT. The initial and final COF values are approximately equal to 0.7. The run-in-periods of 10,000 and 15,000 cycles were found for the tests at 600 and 800 °C, respectively. The COF value of 0.9 was highest during the running-in period in the test at 600 °C. Finally, at the end of the test at 600 °C, the COF value is similar to the one observed at RT. At the highest temperature 800 °C, the COF value fluctuation during the running-in period follows by the continuous decrease with final stabilization about 0.45. The difference in COF values at the end of the tests (RT and 600 °C in opposite to 800 °C) is due to contact between fcc-$(Al,Cr)_2O_3$ and ball (RT and 600 °C) and $\alpha$-$(Al,Cr)_2O_3$ and ball (800 °C). The COF behavior observed in the present study shows a good correlation with the results of Santecchia et al. [31] found in the sliding tests at room and high temperatures. The noise and peaks can be observed in Figure 2. In the case of test at RT, the noisy period lasts for about 20,000 cycles and in the tests at 600 and 800 °C for 8000 and 15,000 cycles, respectively. The noise and peaks can indicate a seizure at the beginning of the sliding tests in analogy with the results of our previous study on the diamond films deposited on Si wafer [26]. The variations in COF value were observed for all tests in the range of 17,000–37,000 cycles, indicating probably that the AlTiCrN layer was worn out and the contact between alumina layer and ball was formed in the course of the tests.

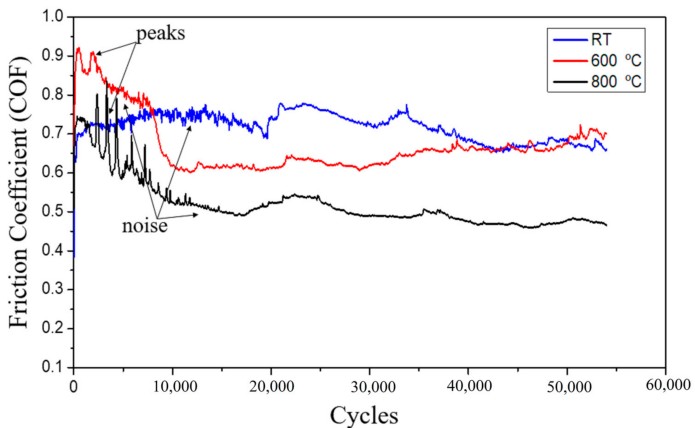

**Figure 2.** COF versus cycles curves taken at RT, 600 and 800 °C.

Figures 3 and 4 show the wear scar depth profiles on the coating and the SEM images taken on the wear scars of coating and balls surface after the tests at RT, 600 and 800 °C. After the test at RT, a relatively smooth wear scar profile can be found in Figure 3a. The wear scar depth is about 0.7 μm, which indicates that at the end of the test a contact was between the ball and $(Al,Cr)_2O_3$ layer, i.e., the second layer ($(Al,Cr)_2O_3$) was partially worn out. Several discontinues grooves were observed on the wear scar surface, see Figure 3b. The width of the wear scar was largest after the test at 600 °C, as well as the presence of grooves within the wear scar was observed (Figure 3e). The roughness of the wear scares surface was highest after the test at 800 °C (Figure 3g,h). The higher roughness of the wear scar after the test at 800 °C indicates significant changes in wear mechanisms as compared with the tests at RT and 600 °C.

Two different areas within the wear scars can be distinguished, which are denoted by I and II Figure 3. The area I is located in the central part and the area II at the border of the wear scar. The area II is shown in more detail in Figure 4. Within this area, the severe wear of the coating was not initiated and only a ball surface plowing was occurred. The analysis suggests that probably ploughing of the outer part of the alumina ball takes place on the hard particles produced during the deposition of the AlTiCrN coating Figure 1a. The comparison of the shape of line scans in (Figure 3a,d,g) with corresponding SEM images in (Figure 3b,e,h) suggest that the plastic deformation of coating occurs in the course of the sliding tests. In spite of the brittle nature of the hard coating, due to probably

relatively small thickness and large area of the coating, the coating can be plastically deformed without cracks formation.

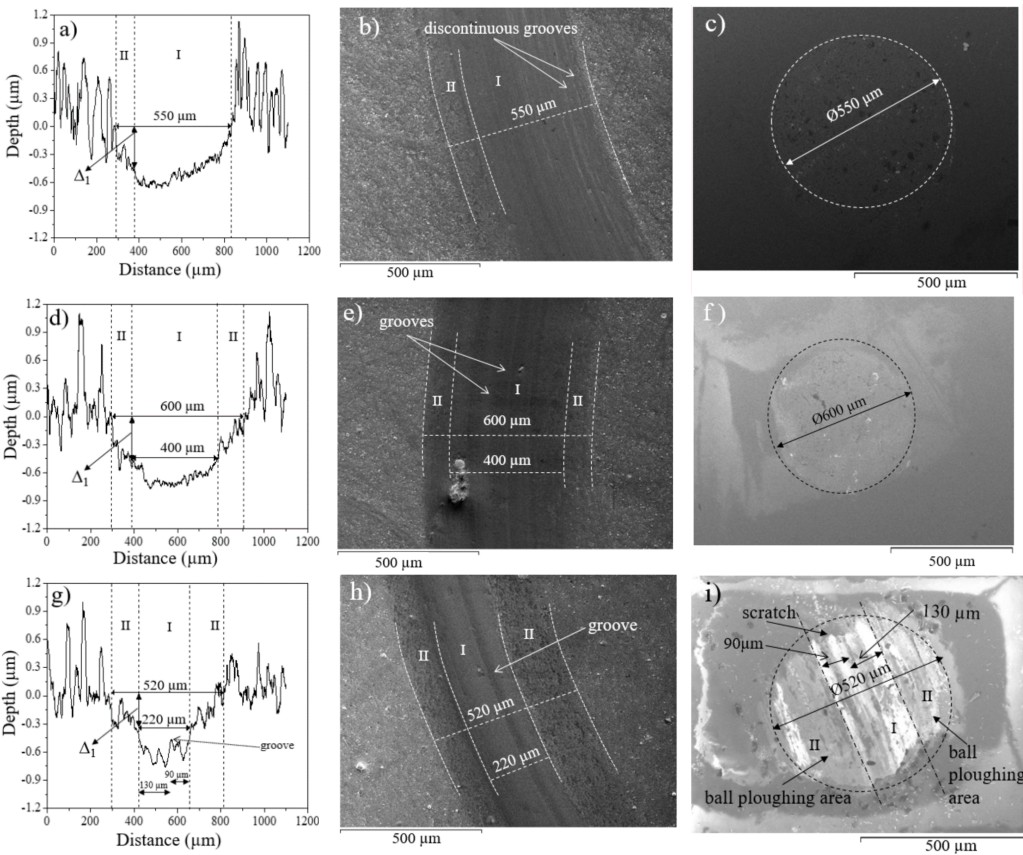

**Figure 3.** Wear scar depth profiles on the coating and alumina balls after sliding tests: (**a–c**) RT, (**d–f**) 600 and (**g–i**) 800 °C.

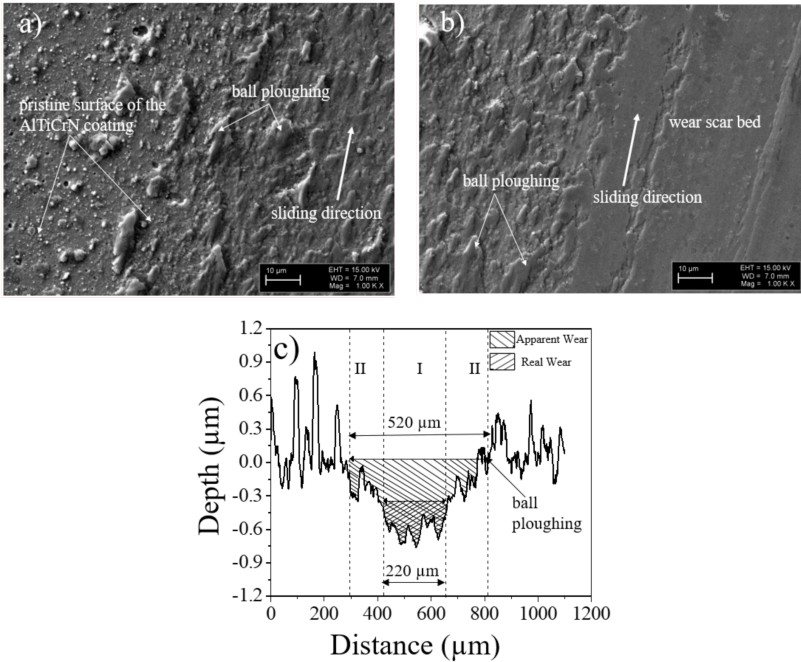

**Figure 4.** SEM images of the sample tested at 800 °C: (**a**) pristine surface and region II, (**b**) regions II and I (see Figure 3g,h) and (**c**) apparent and real wear volumes.

The deformation of the coating was taken into account for the calculation of the wear volume, see Figures 4c and 5. The apparent wear volume should be distinguished from the real one. The apparent wear volume is a sum of the real wear volume and the volume corresponding to the deformation of the coating as shown in Figure 4c. The apparent wear volume was highest after the tests at RT and 600 °C, as well as these wear volumes for both tests are very similar. The smallest apparent wear volume was observed after the test at 800 °C. The real wear volume was smallest for the same test as well. The real wear volume after the test at RT was unable to estimate correctly, therefore it is not shown in Figure 5.

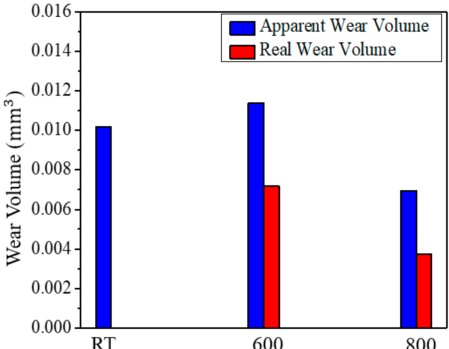

**Figure 5.** Apparent and real wear volumes after the sliding tests at RT, 600 and 800 °C.

Figure 6 shows the EDS data taken on the pristine surface of the coating and within the wear scars after the tests at RT, 600 and 800 °C. The content of oxygen increases, and vice versa the content of nitrogen (presented in AlTiCrN layer) decreases with respect to temperature (Figure 6a). The formation of protective alumina layer can be related with this observation. The EDS data taken within the wear scars on coating shows a presence of a remarkably higher concentration of oxygen, as compared with pristine surface, and the concentration of nitrogen is negligible (Figure 6b). The EDS data allows to conclude that at the end of the tribological tests, the ball was in contact with $(Al,Cr)_2O_3$ layer (Table 1). This conclusion is in good agreement with the deepest valley in line scans in Figure 3, where the depth varied between 0.7–0.75 µm independent of the temperature of the test.

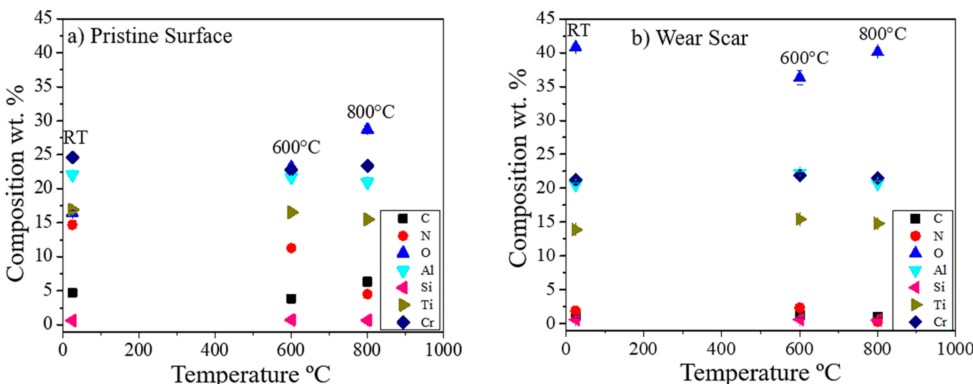

**Figure 6.** EDS data taken on (**a**) the coating pristine surface and (**b**) wear scars surface after the tests at different temperatures.

The nano-scratch tests were performed on the pristine surface of coating after sliding tests at different temperatures. Figure 7 shows the results of the scratch test taken on the coating after the sliding test at 800 °C and the optical micrograph of scratch track. The elastic and plastic deformation and wear of the coating surface can be observed without visible cracks formation. The result shows that the total coating surface deviation during the scratch scan ($\Delta_3 = 0.82$–1.14 µm in Table 3) is relatively high in comparison with the thickness of coating, i.e., 3.6 µm. The deviation due to plastic deformation

and the wear ($\Delta_2$) is the distance between the line scans corresponding to the 1st and 2nd topography scans (Figure 7), i.e., 0.25–0.49 µm (Table 3). The wear is relatively small, thus $\Delta_2$ corresponds mostly to the plastic deformation of the coating, see scratch track in Figure 7. The $\Delta_1$ values shown in Figure 3 correspond to the bending of the area II, i.e., the area where only the ball ploughing was observed. The $\Delta_1$ and $\Delta_2$ values are quite similar due to likely plastic deformation of the coating in both types of tests, i.e., in sliding and nano-scratch tests (Table 3).

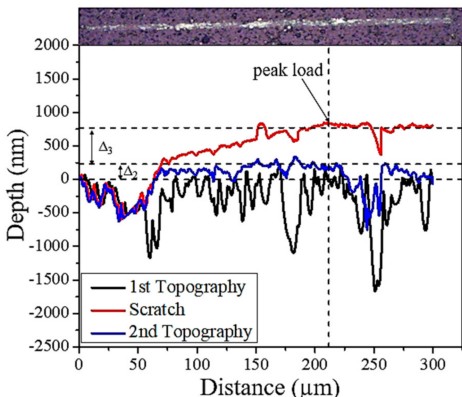

**Figure 7.** Nano-scratch test with scratch track at 400 mN of the peak load taken on the pristine surface of the sample tested at 800 °C in sliding test.

**Table 3.** $\Delta_1$, $\Delta_2$ and $\Delta_3$ values obtained after sliding and nano-scratch tests.

| Samples | $\Delta_1$ (nm) | $\Delta_2$ (nm) | $\Delta_3$ (nm) |
|---|---|---|---|
| RT | 480 | 480 | 1010 |
| 600 °C | 450 | 490 | 1140 |
| 800 °C | 350 | 250 | 820 |

Figure 8 shows the COF value between the coating and the diamond indenter measured for different loads. The COF values (0.12–0.15) observed after the tests at high temperatures correlate with the results of our previous study performed on alumina thin films using the nano-scratch technique with diamond indenter [32]. The low-load (up to the 100 mN) friction behavior is similar independent of the temperature of the sliding tests. It could be probably due to fact that the surface of the coating is relatively rough (Figures 1 and 7) and under the lower loads the COF depends mostly on the surface roughness. In fact, the topography scans are made using small load (0.1 mN). At the higher load there is a difference between the samples tested at RT and at 600 and 800 °C. The difference between COF values can be due to formation of thicker alumina layer on top of the samples tested at 600 and 800 °C, as well as under higher load the stronger adhesion between the diamond indenter and coating surface takes place. This oxide layer can be partially scratched out during the scratch scan. The EDS results confirm that oxide layer is formed on top of AlTiCrN layer at high temperature (Figure 6a).

The mechanical and tribological properties of the coating tested at high temperatures differs from the ones at RT. Formation of the thin oxide layer on the top of the AlTiCrN coating leads to the decrease of COF value between the ball and oxidized AlTiCrN surface in high temperature sliding tests (Figure 2). On the other hand, COF value increases between diamond indenter and oxidized AlTiCrN surface in nano-scratch tests at room temperature (Figure 8). In the sliding tests, the AlTiCrN layer was continuously worn out with following formation of the contact between the alumina ball and $(Al,Cr)_2O_3$ layer (Table 1) in the later stages of sliding. These tests show also the difference between the wear mechanisms of the $(Al,Cr)_2O_3$ layer against alumina ball. The transformation from fcc-$(Al,Cr)_2O_3$ to $\alpha$-$(Al,Cr)_2O_3$ phase takes place at 800 °C. The highest coating deflection was observed on the samples tested at RT and 600 °C, i.e., when there is no phase transformation from fcc to $\alpha$ phase. In the case

of RT and 600 °C, the relatively smooth wear scars surface on the coating and the ball were found, however after the test at 800 °C the surface was rougher (Figure 3). It indicates the change in wear mechanisms at 800 °C, due to probably increase in the hardness of $\alpha$-$(Al,Cr)_2O_3$ and decreasing of the deflection of the whole coating system. The $Al_2O_3$ ball ploughing on the hard AlTiCrN surface peculiarities was found which can result in the noise and peaks in the COF value in sliding tests (Figure 2). In addition, relatively high deformation of the coating was found during the sliding and nano-scratch tests, therefore it suggests that the surface deformation in tribological tests can result in seizure particularly at the early stages of the sliding. Similar phenomena like deflection of the coating, noise and peaks in the COF value and seizure were observed on the diamond films prepared on Si and tested in the sliding tests [26]. Therefore, the effect of deformation of the coating due to high contact pressure between the coating and counterbody can play important role in wear processes.

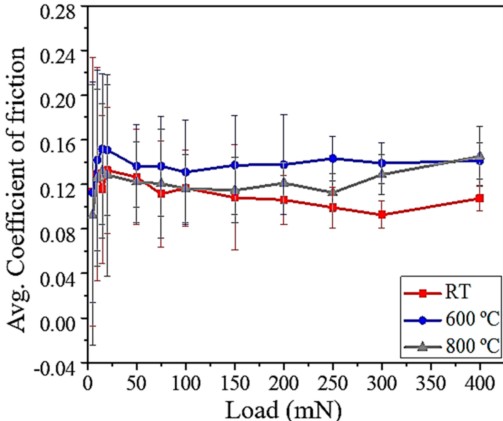

**Figure 8.** Coefficient of friction (COF) versus load measured in nano-scratch tests.

## 4. Conclusions

The mechanical and tribological properties of TiN-AlTiN/nACo-CrN/AlCrN-AlCrO-AlTiCrN coating were evaluated at RT and high temperatures (600 and 800 °C). The increase of wear resistance for coating tested at 800 °C was found due to the formation of the protective oxide layer on top of AlTiCrN and phase transformation within the $(Al,Cr)_2O_3$ layer leading in increase of hardness of this oxide layer and decrease of deflection of the whole coating system. The relatively higher deflection was observed on the samples tested at RT and 600 °C.

The COF value measured in nano-scratch tests between the AlTiCrN layer and diamond indenter is different for the samples heated at different temperatures. Due to formation of the thin oxide layer on the AlTiCrN layer, the COF value is higher for the samples heated at higher temperatures as compare with the sample tested at RT. The coating surface deflection during the scratch test can be significant, i.e., about the 1/3 of the coating thickness at the peak load of 400 mN.

In conclusion, due to high contact pressure between the coating and counterbody, the deformation of coating plays important role in wear processes.

**Author Contributions:** Conceptualization, V.P.; methodology, V.P.; validation, M.Y. and A.B.; formal analysis, A.A. and V.P.; investigation, A.A., M.Y, M.V., and R.T.; resources, V.P.; data curation, A.A.; writing—original draft preparation, A.A.; writing—review and editing, A.A., J.S., A.L., F.S., and V.P.; visualization, A.A.; supervision, V.P.; project administration, V.P.; funding acquisition, V.P. All authors have read and agreed to the published version of the manuscript.

**Funding:** This study was financially supported by the Estonian Ministry of Education and Research under target financing project PUT1369 (Adaptation mechanisms of diamond films in dry sliding wear).

**Conflicts of Interest:** The authors declare no conflict of interest.

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
