# Peer review of "High-Temperature Tribological Performance of Hard Multilayer TiN-AlTiN/nACo-CrN/AlCrN-AlCrO-AlTiCrN Coating Deposited on WC-Co Substrate"

_coatings, doi:10.3390/coatings10090909_

Round 1
Reviewer 1 Report
On the whole I found this an interesting study, particularly the discussion about the importance of the alumina sublayer.
1. The approach to calculating true wear volumes independently of coating plastic deformation appears novel but in figure 3a I couldn't work out why the region II was only observed on the outer side of the wear scar for the RT test? Do you have an explanation?
2. Regarding the correlation between the tribotests and the nanoscratch its also interesting but I would clarify in the caption to Table 3 that you are reporting data from both types of test, otherwise the reader will tend to assume the parameters all relate to the nano-scratching.
3. When you calculate delta 3 in the analysis of figure 7 etc then you will need to remove the elastic contribution of the frame (i.e. the on-load nano-scratch data need to have frame compliance removed for quantitative comparison, just as in nanoindentation). Since it is not mentioned that this has been done I provide the method/reasoning below (apologies if it has been done).
Assuming the frame compliance of the test instrument is of the order of 0.3 nm/mN (fairly typical for this type of machine) then at 400 mN you have around 100 nm of elastic deflection that should be removed from the reported delta 3 values. It is not necessary to do this for the 2nd topography scan as the load is minimal for that. Making the change will not affect any of your conclusions but makes the data more quantitatively accurate.
4. If the nano-scratch data are sufficiently well levelled then the Ra surface roughness measurements from the constant load segments might be worth investigating?
Author Response
Dear Reviewer,
Thank you for your time and effort in reviewing manuscript.
1: The approach to calculating true wear volumes independently of coating plastic deformation appears novel but in figure 3a I couldn't work out why the region II was only observed on the outer side of the wear scar for the RT test? Do you have an explanation?
Indeed, at RT the wear profile seems asymmetrical. We believe that tests conditions strongly influence on the appearance of region II. In this study, we have used relatively strong load (10 N). One could see on the “place” of region II prominent grooves in Fig. 3a. In other words, likely region II was formed, however due to the wear this region was smeared.
2: Regarding the correlation between the tribotests and the nanoscratch its also interesting but I would clarify in the caption to Table 3 that you are reporting data from both types of test, otherwise the reader will tend to assume the parameters all relate to the nano-scratching.
We made changes in this caption. The ∆1 and ∆2 values are quite similar due to likely plastic deformation of the coating in both types of tests, i.e. in sliding and nano-scratch tests (Table 3).
Point 3: When you calculate delta 3 in the analysis of figure 7 etc then you will need to remove the elastic contribution of the frame (i.e. the on-load nano-scratch data need to have frame compliance removed for quantitative comparison, just as in nanoindentation). Since it is not mentioned that this has been done I provide the method/reasoning below (apologies if it has been done).
Assuming the frame compliance of the test instrument is of the order of 0.3 nm/mN (fairly typical for this type of machine) then at 400 mN you have around 100 nm of elastic deflection that should be removed from the reported delta 3 values. It is not necessary to do this for the 2nd topography scan as the load is minimal for that. Making the change will not affect any of your conclusions but makes the data more quantitatively accurate.
Thank you for this notification. For our equipment the frame compliance is 0.33 nm/mN, and influence of this parameter on the final result was automatically removed from data by equipment software.
4: If the nano-scratch data are sufficiently well levelled then the Ra surface roughness measurements from the constant load segments might be worth investigating?
Thank you for suggestion, indeed in this study we did not focus on the surface roughness measurement. In fact, roughness also could influence on the deformation, however it needs more systematic study using samples with different roughness.
Sincerely
Asad Alamgir
Reviewer 2 Report
Manuscript ID: coatings-942704
High-Temperature Tribological Performance of Hard Multilayer TiN-AlTiN/nACo-CrN/AlCrN-AlCrO-AlTiCrN Coating Deposited on WC-Co Substrate
1. The manuscript present an interesting topic. It analyze the mechanical and tribological properties of a multilayer coating at elevated temperatures. For this, ball-on-disk and nano-scratch tests are performed and the results are discussed based on the microstructural and chemical characterization by SEM and X-ray diffraction of the coating. The experimental methodology used is described in detail and facilitates the understanding of the results presented in the paper. The results are explained clearly, so that the reader's compression is facilitated. The conclusions of the work are well summarized and supported by the results presented in the paper.
2. What is the main question addressed by the research? As noted in the comments to the editors, the authors discuss the mechanical and tribological properties at elevated temperatures of a new multilayer coating. This type of multilayer coatings are of great interest in different practical applications, both at room temperature and at elevated temperatures. For this reason, the development and characterization of new coatings of this type is a topic of interest, both from a scientific and industrial point of view.
3. Is it relevant and interesting? In my opinion, the work is interesting as it presents the tribological behavior of a new multilayer coating where the selection of the different layers has been made based on the properties characterized by other authors for the different layers used. This makes interesting a good characterization of the properties of the developed coating. It is true that the authors present a very limited study of the properties of the coating and, therefore, it is not a very relevant work. Logically, a more in-depth study would be interesting, but the results obtained are interesting and can be a good starting point for future studies.
4. How original is the topic? As stated in the report, the originality of the work is average. Numerous works on multilayer coatings can be found in the literature. Therefore, from that point of view it is not a particularly original work.
5. What does it add to the subject area compared with other published material? The contribution of the work is not significant compared to other works in the literature. It is true that, in my opinion, the paper is interesting and I have read the work with pleasure, but the results do not represent a significant contribution to the topic. The experimental approach is rigorously carried out and the results are well explained and properly discussed. However, it is perhaps too simple an approach and the results are as expected.
6. Is the paper well written? In my opinion, yes
7. Is the text clear and easy to read? The authors provide a very good and comprehensive summary about the state of the art of the studied topic, highlight the most important researches. The experimental methodology is explained in detail and facilitates the understanding of the results presented in the paper.The results are presented clearly and are discussed with reference to other works in the bibliography. Therefore, in my opinion the work is clear and easy to understand.
8. Are the conclusions consistent with the evidence and arguments presented? As noted in the comments to the editors, the conclusions of the work are well summarized and supported by the results presented in the paper.
9. Do they address the main question posed? The main question posed in the article is, perhaps, too generic (the mechanical and tribological properties at elevated temperatures of a new multilayer coating). From that point of view, it can be said that they do address the main question posed, but partially since a more in-depth work on the characteristics of the multilayer coating developed would be of interest.
Author Response
Dear Reviewer,
Thank you for your time and effort in reviewing manuscript.
1:The manuscript present an interesting topic. It analyze the mechanical and tribological properties of a multilayer coating at elevated temperatures. For this, ball-on-disk and nano-scratch tests are performed and the results are discussed based on the microstructural and chemical characterization by SEM and X-ray diffraction of the coating. The experimental methodology used is described in detail and facilitates the understanding of the results presented in the paper. The results are explained clearly, so that the reader's compression is facilitated. The conclusions of the work are well summarized and supported by the results presented in the paper.
Thank you for praising our work.
2: What is the main question addressed by the research? As noted in the comments to the editors, the authors discuss the mechanical and tribological properties at elevated temperatures of a new multilayer coating. This type of multilayer coatings are of great interest in different practical applications, both at room temperature and at elevated temperatures. For this reason, the development and characterization of new coatings of this type is a topic of interest, both from a scientific and industrial point of view.
Absolutely agree these coating have great potential applications for both industrial and scientific community.
3: Is it relevant and interesting? In my opinion, the work is interesting as it presents the tribological behavior of a new multilayer coating where the selection of the different layers has been made based on the properties characterized by other authors for the different layers used. This makes interesting a good characterization of the properties of the developed coating. It is true that the authors present a very limited study of the properties of the coating and, therefore, it is not a very relevant work. Logically, a more in-depth study would be interesting, but the results obtained are interesting and can be a good starting point for future studies.
In previous works, we observed deflection mechanism in diamond films, that were monolayer. For the first time we investigate the multilayer coating and glade to find that the deflection phenomena occurred in these coating as well. In future, we plan to perform more in depth investigations.
4: How original is the topic? As stated in the report, the originality of the work is average. Numerous works on multilayer coatings can be found in the literature. Therefore, from that point of view it is not a particularly original work.
We believe that the most interesting part of the work is the phenomenon of deflection observed on multilayered structure and phase transition within the oxide layer.
5: What does it add to the subject area compared with other published material? The contribution of the work is not significant compared to other works in the literature. It is true that, in my opinion, the paper is interesting and I have read the work with pleasure, but the results do not represent a significant contribution to the topic. The experimental approach is rigorously carried out and the results are well explained and properly discussed. However, it is perhaps too simple an approach and the results are as expected.
For initial understanding of tribological properties and coating deflection mechanism in multilayer system our approach is simple. We would like to use simulation/ modelling to understand better this phenomenon. In fact, we expect that deflection is particularly important to explain the wear mechanisms.
6: Is the paper well written? In my opinion, yes
Thanks for appreciation.
7: Is the text clear and easy to read? The authors provide a very good and comprehensive summary about the state of the art of the studied topic, highlight the most important researches. The experimental methodology is explained in detail and facilitates the understanding of the results presented in the paper. The results are presented clearly and are discussed with reference to other works in the bibliography. Therefore, in my opinion the work is clear and easy to understand.
Thanks for appreciation.
8: Are the conclusions consistent with the evidence and arguments presented? As noted in the comments to the editors, the conclusions of the work are well summarized and supported by the results presented in the paper.
Thanks for appreciation.
9: Do they address the main question posed? The main question posed in the article is, perhaps, too generic (the mechanical and tribological properties at elevated temperatures of a new multilayer coating). From that point of view, it can be said that they do address the main question posed, but partially since a more in-depth work on the characteristics of the multilayer coating developed would be of interest.
Thanks for appreciation, we plan to investigate more in-depth in future.
Sincerely
Asad Alamgir
Reviewer 3 Report
Manuscript sent for review with the title „High-Temperature Tribological Performance of Hard Multilayer TiN-AlTiN/nACo-CrN/AlCrN-AlCrO4 AlTiCrN Coating Deposited on WC-Co Substrate” is a good scientific material. The presented results are particularly significant from the point of view of the basic (cognitive) research of multilayers in terms of structure, morphology, as well as mechanical and tribological properties. Both the subject matter and the methodology of conducting experiments are described very well - the research is conducted in a logical and consistent manner.
The entire manuscript has a very good scientific level. All comments are just improvement, not criticism.
The abstract contains all the most important information - purpose, scope and findings.
Introduction - section has an almost adequate informational level. The Authors properly described the importance of multilayers in terms of the layers wear improvement. The introduction does not mention the effect of the substrate on the abrasive wear and does not consider the method of joining the substrate with the layer (or interlayer). According to the knowledge, both the type of substrate and the type of connection (adhesive, mechanical, etc.) are important in mechanical and tribological tests – it is suggested to develop the literature review.
Materials and Methods/ Results and Discussion - the section is correctly described, but there is no information (or appropriate reference to earlier research or literature) why the WC-Co substrate has been used. Is there any utilitarian and practical aspect of such layers on WC-Co? There is also a lack of information on how the surface of the substrate was prepared (polished, sandblasted, pretreatment e.t.c). In the case of ball-on-disc or scratch tests, it is worth noting the roughness (identification of basic roughness parameters) of the substrate surface.
Questions/remarks:
1. Were the nano-roughness results presented in Table 1 measured on the surface or on the cross-section?
2. At what load nanohardness measurements have been taken?
3. Fig. 5 does not present the AW (apparen wear) valumes for the RT sample - is it intentional? Area II (one-side) is marked in the fig.3a.
4. What is the meaning of the numbers (158 and 159) in Fig. 3a and 3d.
5. Fig.6 - the connection of points with the line, presented on the chart, may be burdened with a very large error - especially in the range of measurements for the temperature of 25 and 600 ºC and especially when there is a tendency to increase or decrease in % wt. composition.
Conclusions drawn correctly.
References selected correctly.
Author Response
Dear Reviewer,
Thank you for your time and effort in reviewing manuscript.
1: Were the nano-roughness results presented in Table 1 measured on the surface or on the cross-section?
Sorry, we did not understand well the question, however we try to answer: in Table 1 the nano hardness measured on the surface of coatings.
2: At what load nanohardness measurements have been taken?
The hardness values were obtained from literature.
* - the hardness values adopted from Platit [32].
** - the hardness values adopted from A. Khatabi et al. [29].
3: Fig. 5 does not present the AW (apparent wear) volumes for the RT sample - is it intentional? Area II (one-side) is marked in the fig.3a.
Indeed, at RT the wear profile seems asymmetrical. We believe that tests conditions strongly influence on the appearance of region II. In this study, we have used relatively strong load (10 N). One could see on the “place” of region II prominent grooves in Fig. 3a. In other words, likely region II was formed, however due to the wear this region was smeared. The real wear volume after the test at RT was unable to estimate correctly, therefore it is not shown in Fig. 5.
4: What is the meaning of the numbers (158 and 159) in Fig. 3a and 3d.
Those are typo errors appeared might be during formatting.
5: Fig.6 - the connection of points with the line, presented on the chart, may be burdened with a very large error - especially in the range of measurements for the temperature of 25 and 600 ºC and especially when there is a tendency to increase or decrease in % wt. composition.
In Fig. 6, we make correction by plotting without lines.
Sincerely
Asad Alamgir